# Learning Multi-Stage Tasks with One Demonstration via Self-Replay

**Norman Di Palo**
The Robot Learning Lab
Imperial College London
United Kingdom
n.di-palo20@imperial.ac.uk

**Edward Jonhs**
The Robot Learning Lab
Imperial College London
United Kingdom
e.johns@imperial.ac.uk

**Abstract:** In this work, we introduce a novel method to learn everyday-like multi-stage tasks from a single human demonstration, without requiring any prior object knowledge. Inspired by the recent Coarse-to-Fine Imitation Learning method, we model imitation learning as a learned object reaching phase followed by an open-loop replay of the demonstrator's actions. We build upon this for multi-stage tasks where, following the human demonstration, the robot can autonomously collect image data for the entire multi-stage task, by reaching the next object in the sequence and then replaying the demonstration, and then repeating in a loop for all stages of the task. We evaluate with real-world experiments on a set of everyday-like multi-stage tasks, which we show that our method can solve from a single demonstration. Videos and supplementary material can be found at this webpage.

**Keywords:** Multi-Stage Imitation Learning, Manipulation

## 1 Introduction

One of the principal, long-term goals of robot learning, is to enable a robot to learn a new skill from human demonstrations in a simple and efficient way, without significant effort from the human, and without them requiring expert knowledge of the underlying algorithm. Imitation Learning techniques have been widely adopted in recent years, but these methods often require a considerable human effort. In Sec. 2, we describe the shortcomings of techniques like Behavioural Cloning, Meta Learning, and Reinforcement Learning, that have hindered their widespread adoptions outside of laboratory settings. Several manipulation tasks involve multiple stages, e.g. grasping and then using a tool (Fig.1), or re-arranging a set of objects. But generally, the more stages a task involves, the more difficult it is to learn [1, 2], further exposing the limitations of the aforementioned techniques.

In this work, we introduce a method that allows a human operator to **teach a robot to solve a new, multi-stage manipulation task, with a single demonstration**, without the need for any prior data or knowledge about the object. Our method

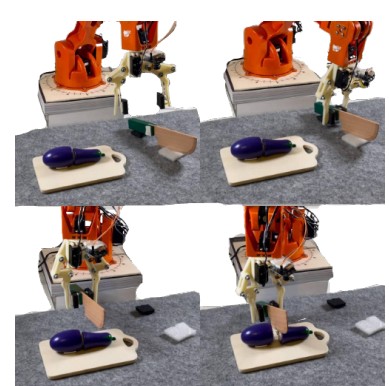

Figure 1: An example of a two stage task: picking up a knife, and then using it to cut a toy vegetable.

is capable of solving a series of everyday-like tasks, and can be effectively used by a non-expert, given its simplicity and need for minimal human effort. The main assumption is that most everyday manipulation tasks can be divided into two phases: a coarse, object reaching phase, and a fine and precise, object interaction phase (Fig. 1, 2). As such, a multi-stage task can be decomposed into several stages, each of which alternates between these two phases. This decomposition forms the basis of our method, which allows the robot to collect all the experience it requires autonomously,

5th Conference on Robot Learning (CoRL 2021), London, UK.

without requiring further human supervision, such as repeated environment resetting. We call our method **Self**-Supervised Learning with Actions **Replay**, or **Self-Replay**. Self-Replay is inspired by the Coarse-to-Fine Imitation Learning method [3], which also models tasks with coarse and fine trajectories, but which can only solve single-stage tasks. Here, we develop a more general method that can learn a wide variety of multi-stage tasks, whilst still only requiring a single human demonstration.

Self-Replay is simple but effective. We evaluated our method on a set of real-world, multi-stage tasks, inspired by everyday primitive skills such as object grasping, object placing, object pushing, shape insertion, and food cutting. Our method is able to solve multi-stage combinations of these skills, whereas in the literature such tasks generally require at least tens of demonstrations, or repeated manual environment resets, or engineered laboratory setups [4, 5, 6, 7, 8]. We also provide a series of ablation studies and comparisons with existing techniques in the literature, to further understand the contribution of each component of our method. Our method only requires an uncalibrated wrist-mounted camera. While the training should take place in an uncluttered environment, we demonstrate how our method can tackle visual distractors at test time.

**Our contributions are the following**: (1) We introduce Self-Replay, a novel self-supervised learning method to solve everyday, multi-stage manipulation tasks from a single human demonstration, with no additional human effort, prior knowledge, or engineering needed. (2) As well as extending it to multi-stage tasks, we extend the data collection method proposed in [3] with a more efficient, active variant, that we call Active Self-Replay. (3) We designed a novel test-time pipeline that allows the robot to tackle multi-stage tasks, whilst also recovering from external disturbances and errors, and being robust to visual distractors. (4) We benchmark our method on a varied set of real-world, everyday-like manipulation tasks, and compare it against other methods from the recent robot learning literature.

## 2 Related Work

The recent Imitation Learning [9, 10, 11, 12] literature has proposed a series of techniques to teach a robot a novel skill, starting from demonstrations, in an efficient way. However, most techniques often require a considerable amount of work, prior knowledge, or supervision from a human operator. Behavioural Cloning methods [13, 4, 14, 5, 7, 6, 1, 15, 16] learn a policy network directly from human demonstrations. While providing demonstrations is conceptually easy, simple tasks may require tens or even hundreds of them, making the process laborious. When dealing with multi-stage tasks, the amount of demonstrations often increases considerably [1, 14]. Meta Learning methods [17, 18, 19, 20, 21] require one or few demonstrations to learn to solve a new task. However, they need large amounts of data collected before hand on similar tasks, another laborious process, and the amount of previously collected data also increases when tackling multi-stage tasks [21]. Plus, they can often only adapt to tasks that are very similar to what was observed during meta-training [17, 18, 19, 20]. Reinforcement Learning methods [2] can use demonstrations to bootstrap the agent's policy [22, 23] or shape its reward function [24, 25]. Even if they require a few demonstration, the autonomous, reinforcement learning phase needs constant human supervision, as the environment need to be reset after each episode. Additionally, a reward function needs to be shaped and provided at each time-step to the agent, which may require additional engineering of the setup [2]. To solve multi-stage tasks, the recent literature has also proposed several methods that combine ideas from different fields: auto-regressive sequence modelling [26], neural task programming [27], exploiting spatial symmetries [28], hierarchical decomposition [1], explicit task planning [29], etc., but each technique requires considerable human effort in terms of either data collection or engineering.

## 3 Method

### 3.1 Background on Coarse-to-Fine Imitation Learning

One of the main concepts our method builds upon is *shifting the imitation learning problem* from learning a model that emulates the demonstration in its entirety, to learning to align the end-effector to the pose where the interaction demonstration started, so that it can repeat the exact actions performed by the operator. The end-effector therefore needs to reach the same relative position to the object that it had at the start of the demonstration. Following this framework, introduced in [3] as

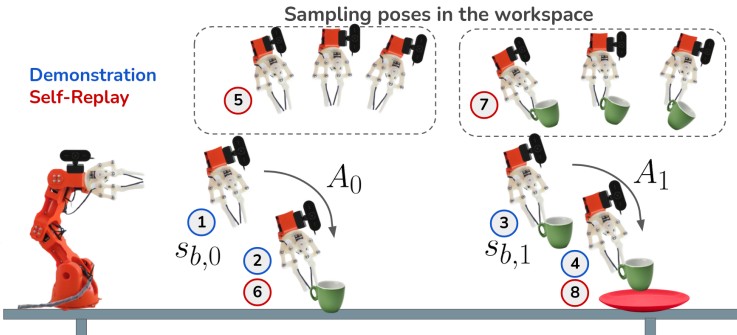

Figure 2: An example of the training steps for a two-stage task: picking a cup and placing it on a plate. The human operator brings the robot to the *bottleneck pose* on top of the cup (1), and records the object interaction phase by grasping the cup and lifting it (2). The operator then brings the robot on top of the plate (3), and records the second object interaction phase (4), by bringing down the cup and releasing it. The operator then resets the environment by placing back the cup. The Self-Replay phase starts: the robot gathers data to learn how to reach $s_{b,0}$ exploring the workspace (5). When the data collection phase ends, the robot moves to $s_{b,0}$ and replicates the first object interaction by executing the recorded actions $A_0$ (6). It then repeats these steps for the second stage (7,8). We suggest to watch the video on our website for a full example.

Coarse-to-Fine Imitation Learning, the human operator provides a demonstration composed of two phases (Fig. 1, 2): they first bring the robot to a pose close to the object, that is defined as the *bottleneck pose*. The bottleneck pose is the pose of the end-effector at which the interaction phase starts. Intuitively, we can imagine it as being fixed in the object's imaginary frame of reference, as it rigidly moves with it. From there, the operator starts the interaction phase, in which they demonstrate how to precisely interact with the object, e.g. picking up a tool, inserting a a plug, etc. (Fig. 1). The assumption, proved effective in [3], is that if the robot can reach the bottleneck pose accurately, hence being in the same relative pose to the object as it was during the demonstration, then replicating exactly the actions recorded during the demonstration is sufficient to correctly interact with the object. The goal of the robot hence becomes to learn to reach the bottleneck pose accurately on novel configurations of the environment, i.e. novel poses of the objects composing the task, without the need to explicitly learn a policy to model the interaction phase.

The bottleneck pose is not an explicit part of the object or of its frame, but is chosen by the operator when providing the demonstration based on how they need to interact with the object. How does the human operator actually choose the bottleneck pose for each stage, separating the reaching and interaction phases? The bottleneck pose is arbitrary, and several choices are equally possible. However, as described in [3], during the interaction phase the actions are replicated in an open-loop manner, and external sources of noise can accumulate an error that is proportional to the length of the interaction phase. Hence, it is suggested to select a bottleneck pose that is sufficiently close to the object.

In a multi-stage task, the operator provides several interaction demonstrations, one for each stage (examples in Fig. 1, 2). Hence, we need to define one bottleneck pose per stage. We denote each bottleneck pose as $s_{b,N}$, where $N$ denotes the $N$-th stage of the task, represented with the spatial coordinates $x_b, y_b, z_b, \theta_b$, where $\theta$ is the rotation around the vertical, $z$-axis. We extract this pose from the robot's kinematics, effectively being a pose in 3D with rotation around the $z$ axis, being it the pose where the demonstration of the interaction phase starts. More details are provided in following sections. We denote $s = \{x, y, z, \theta\}$ as a position and orientation of the end-effector. We define as $o$ the observation, an RGB image that the robot receives from its wrist-mounted camera. Using a wrist-mounted camera, the observations are a function only of the relative pose between the end-effector and the objects. This is a fundamental aspect, as it allows the robot to autonomously gather data that can generalise to novel situations, as described in later sections. Each pose $s$ is computed using the kinematics of the robot. At training time we also have the exact pose of the bottleneck $s_{b,N}$, extracted from the human demonstration, hence we can always reach it precisely through inverse kinematics. At test time this information is unknown and must be deducted from observations. Hence the robot has to learn how to map each RGB observations

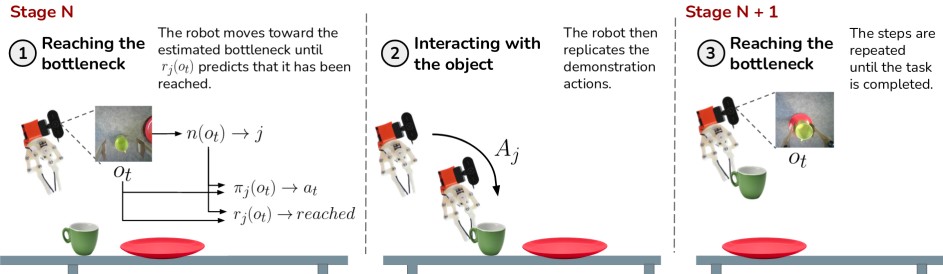

Figure 3: A visual description of the method solving novel configurations of a task at test time. Each stage is solved by first reaching the bottleneck pose, and then replaying the interaction actions. The image is passed through all the networks described in 3.2. The video in the Supplementary Material provides additional examples of this.

$o_t$, where $t$ denotes the current time-step, to action $a_t$ that moves the end-effector closer to $s_{b,N}$ in any novel configuration of the task it needs to solve. The actions are the end-effector spatial velocities, angular velocity around the $z$-axis, and a binary command to open or close the gripper, $v_x, v_y, v_z, v_\theta, c$. All the actions are given in the end-effector frame.

## 3.2 Tackling Multi-Stage Tasks: Overall Architecture

In the previous section we described how Coarse-to-Fine Imitation Learning [3] proposes to solve a novel configuration of a task for which it received a single demonstration: the end-effector must learn to reach the bottleneck pose, the relative pose it had with the object at the beginning of the operator demonstration, and then it can replay the actions executed during the demonstration. To solve a multi-stage task, we need to extend this method to alternate a series of bottleneck reaching and actions replay phases. To do this, we introduce a novel framework of neural networks and a novel self-supervised data collection algorithm, Self-Replay.

To solve a multi-stage task at test time, the robot needs to understand in which stage of the task it is at any time, how to reach the bottleneck of the predicted stage, and if that pose has been reached with sufficient precision, so it can start the interaction phase. All this information must be extracted from an observation, $o_t$ in the form of an RGB image at each time-step $t$, as at test time the absolute pose of the bottleneck is not provided, nor the stage that the robot is currently in.

We designed a framework of three different networks to do so: first, a **stage-recognition network**, $n(o_t) \to j \in [0, N]$, that predicts the stage of the task the robot is currently in, in the form of a discrete class. This prediction conditions the remaining pipeline, as for each stage the bottleneck to reach is different. We predict the current stage at each time-step instead of simply starting from the first and shifting to the next after each interaction phase to be robust to failures and external disturbances. If, for example, the end-effector drops an object during a stage, it needs to realise that it must go back to a previous stage to pick it up again. We then use a **bottleneck reaching network** $\pi(o_t) \to a_t$, that given the current RGB image captured from the wrist-mounted camera, predicts what action to take in the end-effector coordinate frame to reach the bottleneck pose, $s_{b,N}$. Finally, a **bottleneck classification network**, a network that predicts if the bottleneck pose has been reached, $r(o_t) \to \{0, 1\}$. If the network predicts that $s_{b,j}$ has been reached, i.e. is closer than a threshold, the robot executes the series of object interaction actions recorded during the demonstration for the $j$-th stage. The general pipeline is visually described in Fig. 3. Apart for the stage-recognition network, we train a different network for each task stage, and use the stage prediction $j$ to select which one to use. We could have used a single network for each type, adding $j$ to its inputs to condition its output, but we found training separate networks to result in better performance. In the following sections, we describe how Self-Replay can autonomously collect all the data needed to train the aforementioned networks, and hence learn to solve the multi-stage task, starting from a single demonstration.

**Test Time Pipeline** In Fig. 3 we visually describe the pipeline at test time, starting from an RGB observation $o_t$. The networks collaborate to predict (1) in which stage the robot is actually in, hence understanding what bottleneck pose should be reached (2) how the robot should move to approach

the bottleneck of the predicted stage (3) if the bottleneck has been reached, and hence it is time to execute the interaction phase to complete the stage. If the bottleneck is considered non reached, the actions computed by $\pi(o_t)$ are executed by the robot, and then a new observation is obtained. All these estimations are computed from the same input observation $o_t$ and recomputed independently at each time-step.

## 3.3 Providing the Demonstration

The human operator only needs to provide a single demonstration of the task, and then reset the environment as it was before the demonstration. No additional actions are needed from the operator. For each stage of the task, the human operator first brings the robot's end-effector to the object's bottleneck pose $s_{b,N}$ through free-space motion. In this phase, only the final bottleneck pose is recorded. They then demonstrate how to interact with the object moving the robot's end-effector. In this phase, all the actions commanded by the operator are recorded as $A_N = a_{0:T_N,N}$, where $N$ denotes the $N$-th stage. The operator then repeats these steps for all the stages composing the task. An example can be seen in Fig. 2.

The human operator then needs to reset the environment only one time as it was at the beginning of the demonstration, as the object interaction phases may have changed the initial configuration of the objects. This is in strong contrast with other approaches, like reinforcement learning-based methods, that require hundreds or thousands of resets [2], or classic behavioural cloning methods, in which the human operator needs to provide tens of demonstrations [5, 8], resetting the environment each time.

## 3.4 Collecting Data via Self-Replay

After providing a single demonstration of the full task and resetting the environment, bringing the objects to their starting position, all the following data collection and training phases are self-supervised, hence the human operator can disengage and no more human interventions are needed.

As described in Sec. 3.1, 3.2, the agent needs to collect data to train a the models that will allow it to accurately reach the bottleneck pose of each stage from any environment configuration and starting pose. As described before, the use of a wrist-mounted camera makes the RGB observations a function of the relative, and not absolute, pose between end-effector and objects. Hence, moving the end-effector is generally equivalent to moving the object to observe it from different poses. To gather data in a self-supervised fashion we use a method we call Self-Replay.

**Random Self-Replay** Our algorithm extends the method proposed in [3] by being able to autonomously shift between stages to autonomously gather all the data it needs to solve the multi-stage task. We also propose a novel active variant in 3.4. The general approach is to sample a random pose of the end-effector in the workspace, and from there linearly go back to the bottleneck $s_{b,N}$, while recording the observations encountered in the path. The robot explores the area above the bottleneck, to avoid collisions with the objects, and we assume that area to be free from obstacles. More precisely, at each time-step the agent stores data in the form $(o_t, s_{e,t}, s_{b,N}, N)$, where $o_t$ is the image taken from the camera in that pose, $s_e = x_e, y_e, z_e, \theta_e$ is the pose of the end-effector at the current time-step, $s_{b,N}$ is the bottleneck pose, and $N$ denotes the stage of the task we're currently in as an integer. For each pose we can accurately compute the ground-truth relative displacement between the current end-effector pose and the bottleneck, and therefore analytically compute what end-effector movement would bring it closer to the bottleneck. This allows us to then train a model to predict the optimal movement from an RGB observation. This method of data gathering is defined Random Self-Replay, as we randomly sample poses in the workspace to explore different relative poses between end-effector and objects. While randomly sampling poses would be enough, we propose a more efficient and robust active exploration strategy, called Active Self-Replay, that we describe in Sec. 3.4. When the data collection of a task stage is completed, the robot needs to shift to the next stage. To do so, it reaches the bottleneck pose $s_{b,N}$, and then replicates the recorded actions $A_N$ to interact with the object (e.g. moving an object, grabbing an object, etc.). After this phase, the algorithm goes to the next task stage $N \leftarrow N + 1$, repeating the aforementioned steps (Fig. 2), with the next bottleneck pose $s_{b,N+1}$. By autonomously moving between stages, the robot can gather all the data it needs for all the stages without any additional human intervention.

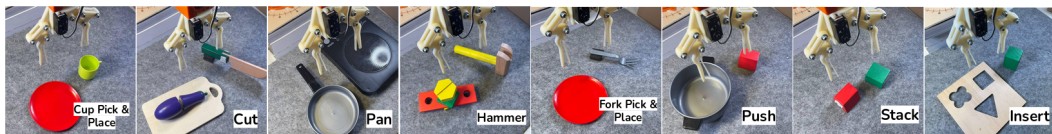

Figure 4: The set of tasks we tested our method on. Additional information can be found in the Supplementary Material and the video on our website.

**Active Self-Replay** We additionally propose an active, self-supervised data collection method. While both methods alternate between collecting data in a self-supervised fashion and then replaying the demonstration's actions to shift to the next stage, the steps of data collection are different. The first stage is identical to the Random Self-Replay phase described before. It collects data repeating the Random Self-Replay cycle $M$ times, gathering an initial dataset. However, differently from Random Self-Replay, we then train our models on this data also during the training phase: after having trained the networks on this initial dataset, the robot will then sample poses from its workspace that maximise the error of the networks, hence areas that have not been explored enough and require more training data. This allows the robot to actively collect more informative data, instead of re-sampling already explored areas. We use a gradient-free sampling based optimiser, and provide more details on the optimisation procedure we implemented in the Supplementary Material.

The data collection and network training steps are repeated until the robot is unable to find a pose in its workspace where the network's error is larger than a threshold $\delta_{err}$, or after a maximum number of steps defined beforehand.

## 4 Experiments

We evaluated our proposed method on a set of real-world, everyday multi-stage manipulation tasks. Based on these experiments, we now answer the following questions: (1) Is our method able to solve a series of common, multi-stage tasks with a single demonstration? (2) Is the Active Self-Replay algorithm more efficient and robust than the Random variant? (3) How does our method compare to similar state-of-the-art methods from the literature? (4) Is reproducing the actions executed by the human operator during object interaction enough to solve the task, or would a learned policy perform better?

Additional experiments are presented in the Supplementary Material, including a comparison with Reinforcement Learning from Demonstrations, a more detailed analysis on the time and sample efficiency of our method compared to other baselines from the literature, a comparison of replaying the operator's demonstration against learning it with a policy, and an investigation of the robustness of our method to distractors. Additional details regarding neural networks and algorithms, and the code for this work, can be found in the Supplementary Material.

### 4.1 Experimental Setting and Tasks

Due to the governmental restrictions during the COVID-19 pandemic, we were unable to conduct experiments in our institution's laboratory. Therefore, we decided to set up a robotics laboratory at home, building a smaller, 6-DOF manipulator with a parallel gripper: the TinkerKit Braccio. Despite the small size of this robot, we were able to conduct all the experiments successfully and answer the above questions, albeit with a reduced workspace. Nevertheless, the results obtained here can effortlessly be replicated on a larger robot. We use a wrist-mounted RGB camera, that captures $800 \times 600$ pixels images, which we resize to $64 \times 64$. We control the robot using a keyboard. Our setup is lightweight and easy to reproduce, only requiring a wrist-mounted RGB camera with no calibration needed. All the networks in this work are convolutional neural networks trained with supervised learning on a single GPU.

**Manipulation Tasks** We designed a set of 8 tasks, inspired by the recent imitation learning literature [3, 13, 4, 5, 7, 8], and by everyday tasks to test our method. We included a variety of common multi-stage human tasks, like picking-and-placing, cutting food, shape insertion, etc. (Fig. 4). Detailed information can be found on our website and Supplementary Material. To generate a test set for the experiments, we instantiate novel configurations of each task by moving the objects in the

Table 1: Percentage of successes of the variants of our method, Random Self-Replay (R. S-R) and Active Self-Replay (A. S-R) and a series of baselines on 20 configurations of each tasks. Experiments are repeated over 3 random seeds and show mean and 1-std.

| Method | Task | | | | | | | |
|--------|------|------|-------|-----|--------|------|------|------|
|        | Cup  | Fork | Stack | Cut | Insert | Push | Pan  | Ham. |
| FlowC.   | 40 ±4.1 | 24 ±2.3 | 13 ±9   | 5 ±4.1  | 6 ±2.3  | 13 ±2.3 | 12 ±2.3 | 18 ±2.3 |
| SIFT     | 8 ±2.3  | 0 ±0    | 0 ±0    | 0 ±0    | 0 ±0    | 0 ±0    | 0 ±0    | 3 ±2.3  |
| LMP-1    | 3 ±2.3  | 0 ±0    | 0 ±0    | 0 ±0    | 0 ±0    | 0 ±0    | 0 ±0    | 0 ±0    |
| RIL-1    | 0 ±0    | 0 ±0    | 0 ±0    | 0 ±0    | 0 ±0    | 3 ±2.3  | 2 ±2.3  | 2 ±2.3  |
| LMP-20   | 62 ±2.3 | 48 ±6.2 | 38 ±6.2 | 23 ±4.7 | 20 ±4.0 | 33 ±2.3 | 50 ±7.0 | 53 ±10. |
| RIL-20   | 60 ±4.0 | 52 ±4.7 | 38 ±4.7 | 17 ±2.3 | 18 ±2.3 | 37 ±2.3 | 48 ±6.2 | 52 ±2.3 |
| DDPGfD   | 3 ± 2.3 | 0 ± 0.  | 0 ± 0.  | 0 ± 0.  | 0 ± 0.  | 0 ± 0.  | 0 ± 0.  | 7 ± 2.3 |
| BC-20    | 30 ± 6.2 | 25 ± 4.0 | 12 ± 2.3 | 7 ± 4.7 | 8 ± 2.3 | 25 ± 7.0 | 28 ± 2.3 | 42 ± 10. |
| R. S-R   | 83 ± 4.7 | 63 ± 4.7 | 58 ± 2.3 | 57 ± 2.3 | 77 ± 4.7 | 63 ± 4.7 | 62 ± 6.2 | 62 ± 2.3 |
| A. S-R   | 88 ± 2.3 | 83 ± 2.3 | 73 ± 4.7 | 73 ± 4.1 | 73 ± 4.7 | 77 ± 4.7 | 73 ± 4.7 | 73 ± 4.7 |

workspace. Position are sampled inside the $30cm \times 20cm$ workspace, and orientation is sampled in a 120 degrees range. The end-effector starts 15 cm above the table. In all the tasks, the robot end-effector can move in 3D space and only rotate around the vertical $z$-axis. For each task, the success criteria is judged by the human operator at the end of the last interaction phase. We provided examples of successes and failures in the Supplementary Material.

## 4.2 Results

**Learning Multi-Stage Tasks from a Single Demonstration** We tested our method by providing a single demonstration of each task. Recording the demonstration takes roughly a minute to the human operator. A video detailing the experiments can be found on our website . We compared Active Self-Replay and Random Self-Replay, (Sec. 3.1) (Table 1). Both variants are trained on the same amount of self-collected data, which is collected autonomously by the robot, following the methods described in 3. We then used the data to train the networks described in Sec. 3.2, keeping the architecture, number of parameters and random weights initialisation fixed on both Active and Random variants. We then set up a test set of 20 random initial configurations of each task by placing the objects randomly in the environment, testing both methods on this same test set and averaging the results over 3 random seeds. At test time, the robot behaves following the method we described in Sec. 3.2 and Fig. 3. In our experiments we observed how, although Random Self-Replay achieves strong performance, it failed on some particularly challenging configurations of the harder tasks, e.g. when the difference in orientation of the object with respect to what observed during the demonstration was large. Indeed, we observe improvements of Active over Random Self-Replay on the harder tasks, like *Cut*, that requires precise orientation of the knife, and *Insert*, which also requires precise orienting the object above the shape (Table 1).

**Comparison with Baselines from the Literature**. We also compared Self-Replay to a series of techniques from the literature, to highlight the improvements of our method. Further details on these experiments can be found in the Supplementary Material. We divide the baselines we developed and compared into three categories:

*1 - One Demonstration Methods:* these methods are the most similar in design to our method, as they only require a single demonstration from the human operator, and no additional prior or posterior work. These methods are FlowControl [30], a recent state-of-the-art technique that used a learned optical-flow model to compute how to align the end-effector, from its current observation, to the bottleneck pose. We also compared a non learning-based keypoint-detection algorithm, SIFT [31], to automatically extract matching features from the current observation and the bottleneck pose observation, used to compute how to align the end-effector to the bottleneck pose.

*2 - Multi-Stage Behavioural Cloning:* we implemented two state-of-the-art methods from the recent literature to learn multi-stage tasks from demonstrations: Latent Motor Plans (LMP) [14] and Relay Imitation Learning (RIL) [1]. Both these methods use human demonstrations to learn a goal-conditioned policy, that can also learn to decompose a longer task into sub-tasks. We also compare these to classic Behavioural Cloning (BC).

*3 - Reinforcement Learning from Demonstrations:* we developed Deep Deterministic Policy Gradient from Demonstrations (DDPGfD) [22, 23], that takes human demonstrations to train a policy, and then uses autonomous Reinforcement Learning to increase its performance. Although the agent autonomously explores the environment during the RL phase, human supervision is still needed to reset the environment after each episode. We use around 30 minutes of exploration, the same time we spend providing demonstrations to BC methods.

We compare our results on a series of everyday-like tasks in Table 1. We provide 1 and 20 demos to the Behavioural Cloning methods. We provide 1 demonstration to DDPGfD, as the human operator also has to spend a considerable amount of time supervising the autonomous exploration phase and resetting the environment. We show that all the baselines fail when receiving only a single demonstration. LMP and RIL perform better than BC, while DDPGfD is unable to solve these tasks consistently. In the Supplementary Material, we also compare the time efficiency of these methods, as motivated by [32], and their sample efficiency. Here, we show that our method is not only more time efficient for the human operator, but can also achieve better performance when using the same number of datapoints than state-of-the-art baselines. Those experiments additionally show that RIL and LMP require around 40-50 demonstrations to reach the performance we obtain with a single demonstration, hence being 40x to 50x more time expensive than Self-Replay.

**Comparison with Behavioural Cloning on an Increasing Number of Stages**

We designed a set of experiments to compare the efficiency and performance of our method on tasks with an increasing number of stages while still receiving only a single demonstration, with respect to Behavioural Cloning (BC), where instead the human operator provides several demonstrations of the task. In particular, we compare it to Relay Imitation Learning (RIL) [1], a recent variation that extends Behavioural Cloning to multi-stage tasks. We designed a stacking task (Fig. 4), where the robot has to stack a varying number of cubes in the correct order, where we can easily add more stages by using more objects.

In our method, we only provided a single demonstration, running Active Self-Replay to autonomously gather data. We compared it to RIL with 10 and 30 demonstrations. For RIL, we provided end-to-end demos of the multi-stage task from 10 or 30 different initial configurations of the environment, recording RGB observations and actions in the end-effector frame in the form of $\{(o_0, a_0), \ldots, (o_T, a_T)\}$. We then trained a policy network $f_{BC}$ with supervised learning on these demonstrations to map $f_{BC}(o_t) \rightarrow a_t$. We used the same network architecture we use as our bottleneck reaching network (Sec. 3.2).

In Table 2, 1 stage corresponds to only grasping the first object, 2 stages to also placing it on a goal $x, y$ position on the table, 4 stages to also place the second object on top of the first. We test the two methods on the same test set of 10 con-

Table 2: Percentage of successes of our method against an end-to-end Behavioural Cloning method, Relay Imitation Learning, with 10 or 30 demonstrations, over 20 configurations of each task.

| Method | Stages | | |
|--------|--------|----|----|
| | 1 | 2 | 4 |
| **Ours** | 100 | 90 | 80 |
| **RIL-10** | 60 | 40 | 15 |
| **RIL-30** | 90 | 75 | 35 |

figurations of the task, measuring the number of successes. We show how Behavioural Cloning, extended to tackle multi-stage tasks with the Relay Imitation Learning method, quickly starts to degrade in performance when the number of stages increases, while our method always obtains a large number of successes.

## 5   Conclusions

In this work, we presented a novel method to learn to solve multi-stage, everyday-like manipulation tasks from a single human demonstration. We build on the foundation of [3], which introduced the concept of Coarse-to-Fine Imitation Learning: we introduce both a novel learning method, Self-Replay, and a framework of neural networks to tackle multi-stage tasks. We empirically demonstrated how our method can solve a wide range of tasks, often used in the literature as benchmarks, while requiring minimal human work, no prior knowledge, engineering, or modelling of the objects to manipulate.

**Acknowledgments**

This work was supported by the Royal Academy of Engineering under the Research Fellowship scheme. We wish to thank the reviewers who provided insightful comments that allowed us to improve the paper significantly. We also wish to thank Vitalis Vosylius for his feedback on an initial draft of the paper.

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
