# OpenReview forum: "Learning Multi-Stage Tasks with One Demonstration via Self-Replay"
_robot-learning.org/CoRL/2021/Conference — CoRL2021 Poster_

### Official Review · Reviewer_yNGA · 2021-07-22

**Originality:** Good
**Technical Quality:** Very Good
**Clarity Of Presentation:** Very Good
**Impact:** 3

**Recommendation:**

Weak Accept: I recommend accepting the paper, but will not argue for my recommendation if the majority of other reviewers have a different opinion.

**Summary:**

This paper extends previous work [1], to effectively solve multi-stage manipulation tasks from a single demonstration. The human expert provides a single kinesthetic task demonstration and decomposes it into bottleneck frames. The robot reaches bottleneck frames, replays the transformed demo at that frame, moves onto the next bottleneck, and repeats until the task is over. The real meat of this method is in the self-supervised tricks and active data collection required to  make error free bottleneck predictions from vision. This process requires minimal human effort and clearly yields huge dividends: the proposed work is far better than behavior cloning at multistage tasks despite using fewer demonstrations.


[1] Johns, Edward. "Coarse-to-Fine Imitation Learning: Robot Manipulation from a Single Demonstration." arXiv preprint arXiv:2105.06411 (2021).

**Issues:**

* I'd add more ablations to show how your method compares vs more naive implementations. For example, what would happen if you omitted the "stage transition" in random self-replay and just predicted where all the bottlenecks were? This is a simpler method but would likely fail due to object occlusions that arise after the first stage. Seeing these comparisons would be helpful.
* Related to prior point - it's hard to tell if Active Self-Replay is really helping over the more naïve variant. One explanation is that the workspace (from video) is too limited to make difference seem apparent. Testing on more configurations/in wider workspace could make this difference more significant.
* This work does not compare against many prior imitation learning baselines. After all, there are prior network architectures and methods (e.g. [2,3]) that improves on behavior cloning for more complicated multi-stage tasks. Thus, more baselines would be helpful.

[2] Lynch, Corey, et al. "Learning latent plans from play." Conference on Robot Learning. PMLR, 2020.

[3] Yu, Tianhe, et al. "One-shot hierarchical imitation learning of compound visuomotor tasks." arXiv preprint arXiv:1810.11043 (2018).

**Reviewer Expertise:**

Very good: Comprehensive knowledge of the area

**Strengths And Weaknesses:**

**Strengths**
* The paper is well written and convincingly explores the design decisions needed to make this method work in practice. Specifically, the improvements to random self replay and active components are sensible ways to boost performance.
* The experiments consider a great deal of tasks and strongly support the hypothesis. The inclusion of robot results despite COVID is a big plus.


**Weaknesses**
* The paper could be seen as a relatively straightforward extension of prior work. I believe the added components are significant enough, but would like to see more thorough ablations done.

**Summary Of Recommendation:**

This paper proposes a non-trivial suite of self-supervised methods in order to extend Coarse-to-Fine imitation learning [1] to multi-stage tasks. By definition this work is incremental, but thorough robot experiments help make up for that. This is just enough to warrant acceptance. However, I should stress that more ablations and baselines could make the work significantly stronger.

---

> ### Author Response · Authors · 2021-08-27
> **Response by Authors to Review of Paper51 by Reviewer yNGA**
>
> > The paper could be seen as a relatively straightforward extension of prior work. I believe the added components are significant enough, but would like to see more thorough ablations done.
>
> We added a new set of experiments during the review period, including new ablations. In particular, we compared Self-Replay with **additional state-of-the-art imitation learning methods** (Table 1, page 7 of the updated paper). The new experimental setting tests all methods’ performance on 20 configurations of the environment and 3 random seeds. We also investigated its robustness to visual distractors during this review period (Table 3, page 13 in the Supplementary Material https://bit.ly/3ksLAYo), and showed the evolution of the performance of all the methods as a function of the operator's time and dataset size (Figure 6, 7 in the Supplementary Material, pages 12 and 13). We also added a new video to our website (https://sites.google.com/view/self-replay) , showing our method's ability to ignore visual distractors autonomously. We believe these experiments, added to the ones already presented in the paper, present a strong series of results that convincingly show the strengths of our method with respect to the baselines.
>
> > I'd add more ablations to show how your method compares vs more naive implementations. For example, what would happen if you omitted the "stage transition" in random self-replay and just predicted where all the bottlenecks were? This is a simpler method but would likely fail due to object occlusions that arise after the first stage. Seeing these comparisons would be helpful.
>
> We tried different variations of our method before and found our current one to be the best performing one. We also tested using a single, bottleneck-conditioned policy to reach the various states, similar to what the reviewer described, but its results were sub-optimal. We did not perform a complete set of experiments to include it as an ablation. We preferred to add more state-of-the-art baselines and the additional experiments we describe in the general response here on OpenReview, which we invite the reviewer to read, e.g. an analysis of the method’s **robustness to visual distractors**, or a comparison of the **time and sample efficiency** of Self-Replay and the baselines.
>
> > Related to prior point - it's hard to tell if Active Self-Replay is really helping over the more naïve variant. One explanation is that the workspace (from video) is too limited to make difference seem apparent. Testing on more configurations/in wider workspace could make this difference more significant.
>
> The intuition is correct: with our current robot, we are limited to a smaller workspace that reduces the advantages of Active Self-Replay. But as the reviewer suggested, we have since extended the experiments comparing Active and Random Self-Replay, to 20 configurations for each task instead of 10. The training and testing are repeated from scratch for 3 random seeds for each task. We show the mean and standard deviation of the variants' performance in Table 1 of the updated paper, and we believe the results are now more convincing.
>
> > This work does not compare against many prior imitation learning baselines. After all, there are prior network architectures and methods (e.g. [2,3]) that improves on behavior cloning for more complicated multi-stage tasks. Thus, more baselines would be helpful.
>
> Since the reviews, we **extended the experiments section by adding additional baselines from the recent literature**, including two state-of-the-art multi-task Behavioural Cloning methods, namely *Latent Motor Plans* from *Learning Latent Plans from Play* [1] (as suggested the review), and *Relay Imitation Learning* [2]. The results can be found in the updated paper, in Table 1.
>
>
> [1] Lynch, Corey, et al. "Learning latent plans from play." Conference on Robot Learning. PMLR, 2020.
>
> [2] Abhishek Gupta, Vikash Kumar, Corey Lynch, Sergey Levine, Karol Hausman. Relay Policy Learning: Solving Long-Horizon Tasks via Imitation and Reinforcement Learning

---

### Official Review · Reviewer_qL1E · 2021-07-23

**Originality:** Fair
**Technical Quality:** Fair
**Clarity Of Presentation:** Fair
**Impact:** 2

**Recommendation:**

Weak Accept: I recommend accepting the paper, but will not argue for my recommendation if the majority of other reviewers have a different opinion.

**Summary:**

* This work extends "coarse to fine" imitation learning to the multi-stage setting.
* Introduces a way to scale up demonstration collection for multi-stage tasks with the following structure: 1) "move to bottleneck 1" phase, e.g. move in free space to pregrasp of an object, 2) "bottleneck 1" replayable open-loop object interaction, e.g. picking up a fork, 3) "move to bottleneck 2", e.g. carry an object above a plate, 4) "bottleneck 2" replayable open-loop object interaction, e.g. place the fork on a plate.
* Turns one demonstration for a particular object configuration into many (for the same particular object configuration) using the following scheme: autonomously reach the first bottleneck pose from many different positions, replay the bottleneck 1 actions, autonomously approach bottleneck two from a variety of poses, replay bottleneck 2 actions.


**Issues:**

* The paper would benefit from clarifying the actual sources of task variation that this work addresses. Clearly state what sources of variation it does not address. The claims that this solves "everyday" tasks should be softened substantially if it does not indeed address everyday sources of task variation.
* The paper would additionally benefit from clarifying the strict dependencies of this method early on (wrist-mounted camera, uncluttered scene).
* Claims of statistical significance should not be made without experiments that support those statements. Without rigorous experiments, it is difficult to assess the quality of the proposed method.

**Reviewer Expertise:**

Excellent: Expert knowledge on the topic of the paper

**Strengths And Weaknesses:**

# Strengths
* Well motivated problem setting: collecting demonstrations for multistage tasks is cumbersome and time-intensive.
* This work introduces a clever and scalable way to collect many demonstrations from only one, assuming certain (fairly strict) requirements about the tasks and environment are met. I can imagine some important commercial applications that meet the requirements of this method.
* Clear description of the method and assumptions around task structure.
* Compelling qualitative results on a real robot.

# Weaknesses
* **Introducing strict assumptions, but not adequately discussing them.** A significant limitation of the setup is that the set of automatically created demonstrations are only valid for one particular object configuration. The paper addresses this by introducing a different assumption: that the agent perceives the scene only through a wrist-mounted camera. The wrist-mounted camera "makes the RGB observations a function of the relative, and not absolute, pose between end-effector and objects. Hence, moving the end-effector is generally equivalent to moving the object to observe it from different poses."  This is furthermore only true when there is no clutter or visual variation in the background. If there is, then the RGB observations of even the wrist camera depend on the absolute position of the scene and the particular configuration of the background. This assumption is never stated but it is nevertheless important. Requiring a wrist-mounted camera (and only a wrist-mounted camera) as well as an uncluttered scene are rigid assumptions. The issue is, without these assumptions, the autonomous data collection becomes far less useful as it is indexed on a particular object configuration---changing the object configuration even slightly requires recollecting all new data.  It's fine to restrict attention to the setting of "wrist-mounted visual manipulation in uncluttered scenes", but this set of assumptions should be very clearly stated early on in the introduction. L124 in Section 3 mentions how important the wrist-camera is to the method: "[the wrist-mounted camera] is a fundamental aspect, as it allows the robot to autonomously gather data that can generalise to novel situations, as described in later sections". The intro should be significantly reworded to accurately convey these (significant) limitations to the reader.

* **Insufficient discussion of the other important sources of variation in imitation**. It is true that imitation learning requires a lot of human effort, and it is a good goal to reduce this effort. But this paper does not appear to address the important sources of effort in behavioral cloning. It provides a way to collect a very large number of demonstrations for one particular scene configuration, covering different initial poses and different trajectories between bottleneck states. Covering pose variation is important, but only *one* of the many kinds of variation that necessitate a large number of demonstrations when doing behavioral cloning. Especially in the "everyday" setting this work claims to operate in, multiple demonstrations are needed to cover changes in the initial positions of the objects, changes in lighting conditions, changes in background, presence of distractor objects etc. If any of these change, all the data automatically collected by this method seems just as brittle as a large set of manually collected demonstrations from a single world state. This is important missing context. Without this context, statements like "...we introduce a novel method to learn everyday-like multistage tasks from a single human demonstration" seem quite strong, as it does not appear to actually improve upon the most important sources of "everyday" variation.


* **No limitations discussed on just how coarse the "coarse" phase can be**. The paper states that it can automatically expand a demonstration set with "without requiring any prior object knowledge". It is hard to imagine how this is true for, say, a cup of coffee, a bowl of candy, a plate with food on it, etc, without requiring object-specific specific scripting like “the container needs to remain upright". Clearly how you script the automatic "move to random poses" phase of collection depends on the object you are dealing with and the scene you are operating in. Otherwise the robot may do unsafe things like collide with objects. Language like "without requiring any prior object knowledge" (which appears multiple times in the paper) should be significantly toned down or the assumptions about which objects this method pertains to should be made more explicit.

* **Strong conclusions from weak experiments.** Experiments are reported without standard deviation over multiple training seeds. Without knowing the standard deviation of the success of the methods, how can we tell if we are observing an actual improvement or just sampling error? "Results therefore show that learning the interaction phase from demonstrations does not provide any *statistically significant* benefit" (emphasis mine). The experiments do not currently carry enough information to assess statistical significance. This is also quite a broad conclusion given the major assumptions around the task setup.  "In our method, we only provided a single demonstration, running Active Self-Replay for around 40 minutes to autonomously gather data. We compared it to BC with 10 and 30 demonstrations." How many minutes of data were the 10 and 30 demonstrations? Was your method simply trained on more data? Without knowing this it's difficult to assess the benefit of the method.

**Summary Of Recommendation:**

UPDATE:
In light of the following, I'm updating my score:
* The authors updated the paper to clearly state the strict dependencies of the method upfront (uncluttered scenes at training time, wrist mounted camera).
* They added new experiments showing that despite these assumptions, the method handles novel configurations of the task even in the presence of unseen visual distractors.
* They added seeded real world evaluations that help communicate the significance of the quantitative results.

---

> ### Author Response · Authors · 2021-08-27
> **Response by Authors to Review of Paper51 by Reviewer qL1E (2/2)**
>
>
> >Insufficient discussion of the other important sources of variation in imitation [...]
>
> Even by experiencing only a single initial pose of the objects, our method is **able to solve the tasks with novel, unseen initial positions of the objects**, as shown in our experiments (section 4.2 and Table 1, page 7 in the updated paper) and videos (https://sites.google.com/view/self-replay) . However, aside from variation to object positions, this review raises further important points: is the method robust to visual changes, such as lighting changes, visual distractors, etc.? To address this, we conducted further experiments during the review period to test the ability of our method to **solve tasks in the presence of visual distractors**, i.e. objects not seen during training. (Supplementary material, page 13)
> As mentioned before, we included a novel set of experiments in the Supplementary Material (https://bit.ly/3ksLAYo) and additional videos on our website (https://sites.google.com/view/self-replay), showcasing the robustness of our method to several visual changes not seen during training. This robustness mostly comes from the strong regularizing ability of Deep Spatial Autoencoders.
> We believe these experiments consider a wide range of possible real-world situations, among different initial positions of objects, distractors, and lighting changes. As most works in the field of imitation learning, such as the influential papers [1], [2], [3], [4], we assume that the environment doesn’t change considerably besides these aforementioned variations.  Stronger variations are considered in Sim-to-Real papers, like [5], although these require models of the robot, the environment, and objects.
> In conclusion, our updated set of experiments show that our method is robust to a range of variations expected in everyday scenarios, including object poses, illumination changes, and distractor objects.
>
> [1] Sergey Levine, Chelsea Finn, Trevor Darrell, Pieter Abbeel, End-to-End Training of Deep Visuomotor Policies
>
> [2] Tianhao Zhang, Zoe McCarthy, Owen Jow, Dennis Lee, Xi Chen, Ken Goldberg, Pieter Abbeel, Deep Imitation Learning for Complex Manipulation Tasks from Virtual Reality
>
> [3] Corey Lynch, Mohi Khansari, Ted Xiao, Vikash Kumar, Jonathan Tompson, Sergey Levine, Pierre Sermanet, Learning Latent Plans from Play
>
> [4] Chelsea Finn, Tianhe Yu, Tianhao Zhang, Pieter Abbeel, Sergey Levine, One-Shot Visual Imitation Learning via Meta-Learning
>
> [5] S James, AJ Davison, E Johns, Transferring End-to-End Visuomotor Control from Simulation to Real World for a Multi-Stage Task
>
>
> >No limitations discussed on just how coarse the "coarse" phase can be. [...]
>
> With "no need for prior object knowledge", we refer to either markers, 3D models of the object, or prior manipulation experience and data collected by the operator. By selecting the bottleneck pose above the object and only exploring the area above that (assuming the robot will always start above the bottleneck pose, which is always the case in all our tasks, and as almost always happens for tabletop manipulation tasks), the robot never collides with objects. We agree that, if the environment had large obstacles, they would need additional care, but it is outside the scope of our work.
>
> For the first stage of a task, the human operator demonstrates how to grasp an object, and the robot emulates those same actions both during training and testing for the fine manipulation part. During the self-supervised data collection phase for the second stage of the task, the object in the gripper then remains in the same pose that the human operator demonstrated. Hence, if a cup is grasped in its upright pose during the demonstration, it will continue to stay in that upright pose during both training and testing. But the robot does not need to understand the concept of “upright” or have any prior knowledge about the cup, as this pose is obtained through the demonstration alone.
>
> > Strong conclusions from weak experiments.
>
> We conducted **additional experiments during the review period** to address the points raised in the reviews, and extended the existing settings, testing our method on more initial configurations of the environment (20 instead of 10)  and averaged of 3 random seeds. We added additional state-of-the-art multi-stage imitation learning baselines, and compare them with our method. We now report the mean and standard deviation of the task solving performance of our method and the baselines in Table 1, page 7, in the updated version of the paper. We additionally added novel experiments in the Supplementary Material (https://bit.ly/3ksLAYo) showcasing how task solving performance increases as a function of operator's time and dataset size (Figures 6, 7, pages 12, 13), tested over different random seeds. We believe this makes the results statistically stronger and more convincing.

---

> ### Author Response · Authors · 2021-08-27
> **Response by Authors to Review of Paper51 by Reviewer qL1E (1/2)**
>
> > A significant limitation of the setup is that the set of automatically created demonstrations are only valid for one particular object configuration. [...] This is furthermore only true when there is no clutter or visual variation in the background. If there is, then the RGB observations of even the wrist camera depend on the absolute position of the scene and the particular configuration of the background. This assumption is never stated but it is nevertheless important. Requiring a wrist-mounted camera (and only a wrist-mounted camera) as well as an uncluttered scene are rigid assumptions. The issue is, without these assumptions, the autonomous data collection becomes far less useful as it is indexed on a particular object configuration---changing the object configuration even slightly requires recollecting all new data.
>
> It is not true that our method requires new data collection for every object configuration. The eye-in-hand setup, with actions relative to the end-effector, provides **invariance to global object poses**, as well as invariance to relative object poses for the objects involved in the different stages of a task. We show in our experiments and videos that even though data collection occurs with the objects in a fixed configuration, the controller **generalises to novel configurations** of objects during testing: we change both their absolute and relative positions at test time, over 20 different configurations, and average results over 3 random seeds. We expanded the experimental section during the review phase, adding more configurations, random seeds, and baselines. The results can be found in the updated paper (Table 1, page 7) and the videos on our website (https://sites.google.com/view/self-replay). They clearly show that the method can tackle novel configurations of the environment, completely different from the configuration experienced during training.
>
> It is true that we assume there is no clutter during the training phase, but the new experiments we conducted during the review phase, as we described in the general response here on OpenReview, show that our method is able to **solve novel configurations of the task even in the presence of unseen visual distractors**. The experiments and results are described in the Supplementary Material (https://bit.ly/3ksLAYo), Table 3, page 13, and in the videos on our website (https://sites.google.com/view/self-replay) . We show how the keypoints-based convolutional neural network, the Deep Spatial Autoencoder, is able to extract keypoints only for the relevant objects in the task, ignoring additional distractors.
>
> > It's fine to restrict attention to the setting of "wrist-mounted visual manipulation in uncluttered scenes", but this set of assumptions should be very clearly stated early on in the introduction. L124 in Section 3 mentions how important the wrist-camera is to the method: "[the wrist-mounted camera] is a fundamental aspect, as it allows the robot to autonomously gather data that can generalize to novel situations, as described in later sections". The intro should be significantly reworded to accurately convey these (significant) limitations to the reader. / The paper would additionally benefit from clarifying the strict dependencies of this method early on (wrist-mounted camera, uncluttered scene).
>
> We indeed specify that the wrist-mounted camera is a fundamental aspect in the original version of the paper, and have now updated the introduction to mention it early on (Page 2, line 52-53). Still, we do not consider the need for a wrist-mounted camera a strong constraint or weakness, as it is generally easy to install and does not require any calibration. We now specify in the paper how the training phase assumes an uncluttered environment (Page 2, line 53-54), but the policy is then robust to clutter at test time: this has been shown by the novel experiment we conducted during this review period, which can be found in the Supplementary Material (https://bit.ly/3ksLAYo, Table 3, page 13).

---

### Official Review · Reviewer_ViCo · 2021-07-24

**Originality:** Fair
**Technical Quality:** Good
**Clarity Of Presentation:** Very Good
**Impact:** 3

**Recommendation:**

Weak Accept: I recommend accepting the paper, but will not argue for my recommendation if the majority of other reviewers have a different opinion.

**Summary:**

The authors propose to learn policies from a single demonstration and multiple bottleneck states (identified by the demonstrator). The robot is then reset to random states from which it learns to reach the next bottleneck state. The robot is able to do so on its own by "linearly interpolating from random state" to bottleneck state. They show this strategy allows them to learn policies quickly on a real robot on a number of manipulation tasks.

**Issues:**

1.Comparison with relevant baselines will help the paper.

**Reviewer Expertise:**

Very good: Comprehensive knowledge of the area

**Strengths And Weaknesses:**

Strengths
1) The proposed approach is interesting, modular, and general enough to be applied to many robotics problems.
2) Real-world robot experiments on multiple tasks to validate their approach.
3) Nice ablation study of how behavior cloning succeeds on tasks with less stages but as the stages increases more complicated algorithms are required.

Weakness
1) One of the new contributions of this work has been explored before in [1]. In [1] the authors suggest having checkpoints with which one could reset the environment from which the agent can explore. This is similar to the Self-Replay part of this work where the authors start exploring from pre-defined checkpoints or bottleneck states.

2) The proposed idea seems difficult to scale to multiple demonstrations or handle variations in workspace without manual intervention. For example, the authors train a bottleneck classifier treating this one state as a positive and other states as negative. If at test time the configuration of objects is different it would make it difficult for the bottleneck classifier to work properly.
There is some recent work on unsupervised video representation learning [2] which claims that key events and phases can be discovered in a self-supervised way. The authors might consider employing something like this to get a scalable bottleneck state from multiple demonstrations.

3) Evaluation missing relevant baselines:
The baseline considered in this work is FlowControl which is a policy learning method that uses optical flow. It is not clear how that is the best baseline for the proposed approach that tackles long-horizon tasks. Approaches like [3],[4] or [5] which solve long horizon tasks by breaking it into smaller tasks seem more suited here.

4) How many extra interaction steps is required by the proposed algorithm in the Random SelfPlay stage to achieve good performance? The lower bound seems to be pure behavior cloning where no extra interaction steps is required. It will be good to see trend of that graph, so that we can estimate if better performance can be achieved by more steps in the SelfPlay stage.

References

[1] Ecoffet, Adrien, Joost Huizinga, Joel Lehman, Kenneth O. Stanley, and Jeff Clune. "First return, then explore." Nature 590, no. 7847 (2021): 580-586.

[2]Dwibedi, Debidatta, Yusuf Aytar, Jonathan Tompson, Pierre Sermanet, and Andrew Zisserman. "Temporal cycle-consistency learning." In Proceedings of the IEEE/CVF Conference on Computer Vision and Pattern Recognition, pp. 1801-1810. 2019.

[3] Learning to Generalize Across Long-Horizon Tasks from Human Demonstrations. Ajay Mandlekar, Danfei Xu, Roberto Martín-Martín, Silvio Savarese, Li Fei-Fei

[4] D. Xu, S. Nair, Y. Zhu, J. Gao, A. Garg, L. Fei-Fei, and S. Savarese. Neural task programming: Learning to generalize across hierarchical tasks. In 2018 IEEE International Conference on Robotics and Automation (ICRA), pages 3795–3802. IEEE, 2018.

[5] Abhishek Gupta, Vikash Kumar, Corey Lynch, Sergey Levine, Karol Hausman. Relay Policy Learning: Solving Long-Horizon Tasks via Imitation and Reinforcement Learning

**Summary Of Recommendation:**

While the paper is quite well-written and has real-world experiments for multiple (which are difficult to setup), the method is not scalable, missing some baselines comparisons and has limited technical contribution. It was however interesting to see their proposed approach get good performance for manipulation tasks in the real world. Hence I am not recommending the acceptance of the paper in its current form.

**UPDATE**
I thank the authors for newer experiments and additional analysis. It definitely adds more value to the paper.

It seems one factor that makes their method generalize using few samples is the use of a camera on the wrist. It would be important to show if the method can generalize to new configurations from third-person views or the wrist-view is crucial for success. If it is, it should be reflected in title/abstract.

By scalable I meant addressing the scenario when multiple demonstrations are available? Since it is a leaning based method, there has to be configurations (single demonstrations) that make it difficult to generalize to newer (out of distribution) settings.

However with the newer experiments, I can recommend a weak accept if the authors include limitations of their approach.

---

> ### Author Response · Authors · 2021-08-27
> **Response by Authors to Review of Paper51 by Reviewer ViCo (2/2)**
>
> > Evaluation missing relevant baselines: The baseline considered in this work is FlowControl which is a policy learning method that uses optical flow. It is not clear how that is the best baseline for the proposed approach that tackles long-horizon tasks. Approaches like [3],[4] or [5] which solve long horizon tasks by breaking it into smaller tasks seem more suited here. /Comparison with relevant baselines will help the paper.
>
> As suggested, we have **considerably expanded the experiments section** during the review phase to address the issues raised by the reviewers, both in the main paper and the Supplementary Material (https://bit.ly/3ksLAYo)l, implementing more state-of-the-art multi-task methods (Table 1, page 7 in the updated version of the paper). We initially focused on FlowControl because it is the most similar method to ours, as it is also designed to solve tasks from a single demonstration. Differently from our method, it needs a pre-trained optical flow model, while we can learn to solve the task without pre-trained networks. Regardless, we followed the reviewer’s suggestions, also shared by other reviewers, and added more state-of-the-art baselines, including *Relay Imitation Learning*, paper [5] in the reviewer’s response, and *Latent Motor Plans* from *Learning Latent Plans from Play*. *Neural task programming* [4] is a considerably different approach that learns a planner, and as they specify in their paper, needs to be trained in simulation on thousands of demonstrations, hence being infeasible to train in the real world. We chose [5] as an additional baselines from the reviewer’s proposals due to its strong reported performance and suitability to multi-stage tasks.
>
> > How many extra interaction steps is required by the proposed algorithm in the Random SelfPlay stage to achieve good performance? The lower bound seems to be pure behavior cloning where no extra interaction steps is required. It will be good to see trend of that graph, so that we can estimate if better performance can be achieved by more steps in the SelfPlay stage.
>
> Thank you for this suggestion. We have since provided an **additional analysis of the time and sample efficiency** in the Supplementary material (https://bit.ly/3ksLAYo Figures 6, 7, pages 12, 13), conducted during the review phase. It shows how the performance of the various methods changes as a function of the operator's time spent on the robot and the size of the dataset used. While we already argued that our method is considerably more sample efficient than baselines, needing only one demonstration, we also now show how the Coarse-to-Fine structure makes it more sample efficient than state-of-the-art multi-task behavioural cloning techniques.
>
>
> > [...]  the method is not scalable.
>
> We argue that our method is **considerably more scalable than the baselines**, as also noted by reviewer qL1E and by the Area Chair in their review and meta-review. We trained a robot to solve several multi-stage manipulation tasks with a single demonstration; all the remaining data is collected by the robot fully autonomously.
>
> We designed one of our experiments specifically to show the scalability of our method to an increasing number of stages (Table 2 in the updated version of the paper, page 8). We show how our method can solve tasks with multiple stages with a single demonstration of the full task. In contrast, a state-of-the-art multi-stage Behavioural Cloning method, Relay Imitation Learning [1], requires an increasing number of demonstrations to reliably solve the same task when the number of stages increases, as our experiment show. We conducted additional experiments during the review phase, comparing the time and sample efficiency of our method against the reported baselines. The graphs can be found in the Supplementary Material (https://bit.ly/3ksLAYo), Figure 6 and 7. Those graphs show how our method is up to 40x more time efficient. In addition, we also show that our method is more sample efficient, obtaining a better performance than the baselines given the same amount of training data. In this setting, the policy only has to learn how to reach the bottleneck pose in novel configurations of the environment, without the need to learn the fine, complex manipulation part.

---

> ### Author Response · Authors · 2021-08-27
> **Response by Authors to Review of Paper51 by Reviewer ViCo (1/2)**
>
> > One of the new contributions of this work has been explored before in [1]. In [1] the authors suggest having checkpoints with which one could reset the environment from which the agent can explore.
>
> While there are some similarities in the ideas behind our method and that paper, the settings and the algorithms are fundamentally different. First, [1] is designed to be used in virtual scenarios, where a precise and complete reset of the environment to an arbitrary state is possible: this is impossible in the real world, and would need continual manual resetting of the environment by the human operator, thus requiring a considerable effort. Second, we structure Imitation Learning as a coarse phase where the end-effector reaches a bottleneck pose, followed by a fine manipulation phase where the demonstration is replayed. This structure is considerably different to [1]. [1] is a pure end-to-end Reinforcement Learning method, tested on Atari video games, and performs random exploration on the environment to learn a complete policy that maximises rewards. Our method collects self-supervised data to train a policy that learns to align to the bottleneck pose: we do not use rewards, but use the knowledge of the inverse kinematics to label every observation with the action that would bring the robot closer to the bottleneck. Furthermore, we do not learn the fine manipulation part, but replicate the operator’s actions. Finally, the networks we use are considerably different, being composed of a policy, a stage recognizer, and a bottleneck classifier, instead of an end-to-end policy.
> The tasks we aim to solve are very different to those in [1], we focus on real-world, everyday-like manipulation tasks, for which **using pure Reinforcement Learning would be considerably more time expensive, as we demonstrate in our time-efficiency comparisons** (Supp. Material, https://bit.ly/3ksLAYo, Figure 6, page 12)
>
> > The proposed idea seems difficult to scale to multiple demonstrations or handle variations in workspace without manual intervention. For example, the authors train a bottleneck classifier treating this one state as a positive and other states as negative. If at test time the configuration of objects is different it would make it difficult for the bottleneck classifier to work properly. There is some recent work on unsupervised video representation learning [2] which claims that key events and phases can be discovered in a self-supervised way. The authors might consider employing something like this to get a scalable bottleneck state from multiple demonstrations.
>
> We designed our method to be able to solve tasks from a single demonstration, and as we show in our experiments, **multiple demonstrations are not needed**. This is a strength of our method, not a weakness. It makes our method considerably more time-efficient for the human operator than methods requiring multiple demonstrations, without sacrificing task-solving performance (an analysis of time efficiency was added to the updated version of our Supplementary Material https://bit.ly/3ksLAYo , Figure 6, page 12). In our experiments, that we expanded during this review phase, we show that our method, trained on a single demonstration, **is able to solve the task when the configuration of objects is different in training and testing**, as we show in Table 1 (page 7 of the updated paper) and in our videos, which we encourage the reviewer to watch on our website (https://sites.google.com/view/self-replay). Furthermore, we also added novel experiments during the review phase, and tested our method’s robustness to the **presence of novel distractor objects**, and lighting changes, as we show in the Supplementary Material (https://bit.ly/3ksLAYo, Table 3, page 13) and in the videos on our website. Experiments are repeated over 20 configurations of the environment and averaged over 3 random seeds.
>
> So, whilst methods such as Behavioural Cloning could learn from multiple demonstrations, where each demonstration is given with different object configurations, different distractors, and different lighting conditions, this would require an *extremely large number of demonstrations*, whereas our method achieves all of this from just a single demonstration. As also noted by reviewer tkPW, "*the trained networks are able to generalize to new configurations of the scene due to training on observations collected from a wrist camera, which does not tie the observations to a specific configuration of the objects. This is a pretty neat trick and naturally allows the robot to generalize to new configurations of the scene.*"
>
> Since Self-Replay only requires a single demonstration, we argue that it is more data efficient for the operator to directly select a bottleneck pose in their demonstration, rather than training a model that can discover bottlenecks from multiple demonstrations.

---

> ### Author Response · Authors · 2021-09-06
> **Updated Response by Authors after Updated Review**
>
> We thank the reviewer for their time reading our rebuttal and analyzing the updated version of our paper.
> As suggested in the updated recommendation, we will include an additional discussion of the points raised: the importance of the first person camera, and the difference with respect to a third person camera, and what are some possible limitations of the single demonstration scenario.
> We are glad the reviewer confirmed that the additional experiments and analysis add value to the paper, and we will further discuss these points to add more clarity to the final version.

---

### Official Review · Reviewer_tkPW · 2021-07-24

**Originality:** Good
**Technical Quality:** Good
**Clarity Of Presentation:** Good
**Impact:** 3

**Recommendation:**

Weak Accept: I recommend accepting the paper, but will not argue for my recommendation if the majority of other reviewers have a different opinion.

**Summary:**

This paper tackles multi-stage tasks given a single demonstration. In order to do so, the algorithm is composed a stage-recognition network, which classifies which stage the agent is in; a bottleneck reaching network (one for each stage), which gives a policy to reach the bottleneck state; and a bottleneck classification network (one for each stage), which predicts whether the bottleneck state has been reached. At each bottleneck state, the robot simply replays the actions from the demonstrations to complete the stage. The data to train these networks is collected in a self-supervised manner.

**Issues:**

- It’s noted that training separate policies for each stage leads to better performance, compared to training a single policy conditioned on the task id. What if the policy was conditioned on the bottleneck state? Would this joint training help?

- In the self-supervised data collection phase, presumably the robot can disturb the scene. For example, the cup in the Cup Pick & Place task could be knocked over. Does this not require human intervention to correct the scene?

- How sensitive is the performance to the choice of the bottleneck poses? It’d be useful to see how different bottleneck choices for the same demonstration affects performance.

- It would be helpful to break up Sec. 3.4 by outlining the objectives used to train the networks.

- The link to the website seems to be missing from the paper.

**Reviewer Expertise:**

Good: General knowledge of the area

**Strengths And Weaknesses:**

Strengths:
- This work studies imitation of a multi-stage task from a single demonstration under high-dimensional image observations. This is a challenging setting and is highly relevant to the CoRL community. Additionally, the experiments are conducted on a real robot.

- The proposed approach only requires 30-60 minutes of self-supervised data collection before it learns to successfully perform the demonstrated task. Hence, the only human supervision required is a single demonstration with labels for each stage and the corresponding bottleneck state.

- The trained networks are able to generalize to new configurations of the scene due to training on observations collected from a wrist camera, which does not tie the observations to a specific configuration of the objects. This is a pretty neat trick and naturally allows the robot to generalize to new configurations of the scene.

Weaknesses:
- The experimental evaluation only includes one comparison to prior work on a pair of tasks, so it’s difficult to understand how well this compares to existing systems for this setting. The main text points to the appendix for a comparison to RL from demonstrations, but the results are missing.

- It’s also unclear how significant the results are from just the reported success rates. For example, the difference between Random Self-Replay and Active Self-Replay is just one more successful trial out of ten on some of the tasks. It would be helpful to test on a larger number of configurations.

- This approach assumes that prior to achieving the bottleneck pose for an object, the object remains fixed in the scene. However, it seems like the robot could potentially disturb the object during the self-supervised data collection. Currently, the actions used to reach the bottleneck pose are the difference between the bottleneck and current pose, which would be fine if the direct trajectory to the bottleneck pose is obstacle-free and does not disturb the objects. Otherwise, the robot would likely move the objects and need more human intervention.

**Summary Of Recommendation:**

I’m recommending a weak accept. The overall results are impressive for a difficult problem, but it’s difficult to assess the performance of the approach and the difficulty of the tasks without more comparisons.

---

> ### Author Response · Authors · 2021-08-26
> **Response by Authors to Review of Paper51 by Reviewer tkPW**
>
> > The experimental evaluation only includes one comparison to prior work on a pair of tasks, so it’s difficult to understand how well this compares to existing systems for this setting. The main text points to the appendix for a comparison to RL from demonstrations, but the results are missing.
>
> We considerably **expanded the experiments section** during the review phase, both in the main paper  (Table 1, page 7) and the Supplementary Material (https://bit.ly/3ksLAYo), implementing more **state-of-the-art multi-task Behavioural Cloning methods**, as suggested. We also clarified the results of Reinforcement Learning, moving it to the main paper, while keeping the implementation details in the Supplementary Material for space reasons.
>
> > It’s also unclear how significant the results are from just the reported success rates. For example, the difference between Random Self-Replay and Active Self-Replay is just one more successful trial out of ten on some of the tasks. It would be helpful to test on a larger number of configurations.
>
> We expanded the experimental results during the review phase by both testing on **20 configurations**, instead of 10, and also **averaging the results over 3 random seeds**, where we re-started the training and testing from scratch for each seed. These results are now much more statistically significant. (Table 1, page 7 of the updated paper)
>
> > This approach assumes that prior to achieving the bottleneck pose for an object, the object remains fixed in the scene. However, it seems like the robot could potentially disturb the object during the self-supervised data collection. Currently, the actions used to reach the bottleneck pose are the difference between the bottleneck and current pose, which would be fine if the direct trajectory to the bottleneck pose is obstacle-free and does not disturb the objects. Otherwise, the robot would likely move the objects and need more human intervention.
>
> By selecting the bottleneck pose to be above the object when providing the demonstration, and by letting the robot explore the area above it (assuming the robot will always start above the bottleneck pose), the robot can explore the environment freely without hitting the object, as we also show in our videos (https://sites.google.com/view/self-replay) and tasks. We were able to autonomously gather data in all the 8 tasks without hitting the objects placed on the table simply by selecting the bottleneck above the object of interest. It is true that large obstacles might require additional care, but this is beyond the scope of this work at the moment.
>
> > It’s noted that training separate policies for each stage leads to better performance, compared to training a single policy conditioned on the task id. What if the policy was conditioned on the bottleneck state? Would this joint training help?
>
> We initially tried training a single, goal-conditioned policy, but that method performed worse than the current design. We will still explore this possibility in future work to enable more efficient transfer learning.
>
> >In the self-supervised data collection phase, presumably the robot can disturb the scene. For example, the cup in the Cup Pick & Place task could be knocked over. Does this not require human intervention to correct the scene?
>
> As described above, by selecting the bottleneck pose above the objects we can freely explore the environment without hitting the objects. In all our experiments, contacts never happened.
>
> > How sensitive is the performance to the choice of the bottleneck poses? It’d be useful to see how different bottleneck choices for the same demonstration affects performance.
>
> We have now studied our method over 3 random seeds during training, which results in 3 different bottleneck poses for each of the 8 tasks. When replicating the experiments over random seeds, the bottleneck pose selected by the operator can change slightly. Despite this, we observed the results to be consistent, without being influenced by the choice of the bottleneck pose. (results in Table 1, page 7 of the updated paper)
>
> >The link to the website seems to be missing from the paper.
>
> The link can be found in the abstract, both in the paper and here on OpenReview. Here it is as well: https://sites.google.com/view/self-replay

---

### Author Response · Authors · 2021-08-26
**General Response**

We wish to thank all the reviewers for their detailed and insightful reviews and comments. We followed the reviewers’ suggestions to strengthen the paper with **a series of new and expanded experiments during this review period**. We believe these novel experiments address many of the questions raised in the reviews and offer novel insights into our method and its abilities. We have also written separate replies to each individual reviewer to address their individual points. We have uploaded an updated version of our paper here on OpenReview, and we provide a direct link to the **new Supplementary Material** at https://bit.ly/3ksLAYo, which can also find on our anonymous website: https://sites.google.com/view/self-replay. We also added novel videos to this webpage showcasing the method’s robustness to visual distractors.

The main additions to our updated paper are the following:
- We implemented **two additional state-of-the-art imitation learning** methods and compared them with our method. Namely, we added comparisons to *Latent Motor Plans* from “*Learning Latent Plans from Play*” [1], as suggested by reviewer yNGA, and “*Relay Imitation Learning*” [2], as suggested by reviewer ViCo. These methods are recent extensions of Behavioural Cloning designed to tackle multi-stage tasks. We additionally compared our method to a *SIFT* [3] based keypoint extraction method, and added more details on the comparisons with Reinforcement Learning from Demonstration. These experiments are **detailed in Table 1, page 7 of the updated paper**. The testing phase has been extended to 20 configurations instead of 10. As suggested in the reviews, we **ran our experiments on multiple random seeds** and more test time configurations to provide more statistically significant results. For each task, we repeated the entire training and testing procedures from scratch for 3 random seeds. reporting mean and standard deviation.

- We now provide a more detailed analysis of **time and sample efficiency** in the Supplementary Material (https://bit.ly/3ksLAYo, Figures 6, 7, pages 12, 13). We show that our method only requires less than two minutes of the operator's time to teach a robot a new task, which is an order of magnitude more time-efficient than the baselines. We additionally demonstrate that our method is more sample efficient, and obtains better performance than the baselines using the same dataset size. With the Coarse-to-Fine design, our method *only needs to learn how to reach the bottleneck pose*, without the need to learn the fine, and generally harder, manipulation part. Hence, Self-Replay can learn to solve various tasks using fewer datapoints than the state-of-the-art baselines. (Supplementary Material, Figures 6, 7, pages 12, 13, https://bit.ly/3ksLAYo)

- We now demonstrate our method's **robustness to visual distractors**. As some reviewers commented, it was unclear how the method would perform under visual changes in the environment. We show how the keypoint-based convolutional network we use can ignore visual distractors and only track the objects which are relevant to the task. We provide new videos showing this ability and measured task-solving performance with and without distractors, empirically showing robustness to distractors. (Table 3, page 13 of Supplementary Material (https://bit.ly/3ksLAYo) , videos: https://sites.google.com/view/self-replay)


-[1] Lynch, Corey, et al. "Learning latent plans from play." Conference on Robot Learning. PMLR, 2020.

-[2] Abhishek Gupta, Vikash Kumar, Corey Lynch, Sergey Levine, Karol Hausman. Relay Policy Learning: Solving Long-Horizon Tasks via Imitation and Reinforcement Learning

-[3] David G. Lowe, Distinctive Image Features from Scale-Invariant Keypoints

---

### Meta-Review · Area_Chair_LYc5 · 2021-08-13

**Recommendation:** Accept (Poster)
**Confidence:** 4

**Metareview:**

In this paper, the authors propose a method to solve multi-stage task from one demonstration. Multiple bottleneck states are defined and recognized. It is well recognized that the system is able to collect scalable data in a self-supervised fashion. However, there are some concerns about how to handle multiple demonstrations, some practical situations, etc. Also, more comparison experiments with stronger baselines are lacking.

In the rebuttal session, the authors have responded to the reviewers' concerns and more details and baselines are provided. And the reviewers have reached a consensus of accepting this paper.

---

> ### Public Comment · (anonymous) · 2021-08-30
> **Response to the Meta Review by Area Chair LYc5**
>
> We thank the Reviewers and Area Chair for their detailed reviews and insightful comments, which have helped us to strengthen the paper: during the review period, we **considerably expanded the experiments section**, addressing the points raised by the reviews. A detailed description can be found in our **General Response** here on OpenReview.
> We responded to each review, discussing all the perplexities or problems raised by the reviews. In this particular response, we will address the main points raised in the meta review by the Area Chair.
>
> >How to handle multiple demonstrations [...] / Stronger baselines are lacking.
>
> Our method is specifically designed to be able to **learn to tackle multi-stage tasks from a *single demonstration***. While this may seem strongly different from the general Learning from Demonstrations frameworks, where multiple demonstrations are needed to achieve satisfactory performance, in our paper we include a series of experiments that show how our method can in fact be remarkably more time-efficient without sacrificing performance.
> During this review period, we added a **new series of state-of-the-art Behavioural Cloning baselines** from the recent literature, as suggested by the reviewers. Namely, we added comparisons to *Latent Motor Plans* from *“Learning Latent Plans from Play”* [1], as suggested by reviewer yNGA, and *“Relay Imitation Learning”* [2], as suggested by reviewer ViCo. These methods are recent extensions of Behavioural Cloning designed to tackle multi-stage tasks. We compared these and other baselines, including Reinforcement Learning, to our method, Self-Replay (Table 1, page 7). Moreover, during the review period we ran the experiments on 3 random seeds, averaging the results and showing mean and standard deviation for statistical significance. These results show how our method, receiving a single demonstration, can **outperform the state-of-the-art baselines** while they receive even 20 demonstrations.
>
> We added a new series of experiments to **investigate the time and sample efficiency** of our method and the baselines. We ran a series of experiments, detailed in the Supplementary Material (https://bit.ly/3ksLAYo, Figure 6, 7, page 12. 13), that shows how our method, when receiving a single demonstration, is an **order of magnitude more time efficient** with respect to state-of-the-art Imitation Learning baselines in terms of time spent by the operator on the training process. Moreover, we also demonstrated how our method is **more sample efficient**, i.e. obtaining a better performance given the same dataset size. This is thanks to our Coarse-to-Fine decomposition, that allows our policy to only learn how to align to the bottleneck pose, without the need to learn the complex, fine manipulation phase.
>
> Table 2, page 8, also included in the original version of the paper, demonstrates how our method, receiving a single demonstration, is **more scalable to tasks with multiple stages** than a state-of-the-art Imitation Learning baseline. The latter needs an increasing number of demonstrations when the number of stages increases to still obtain a satisfying performance, while our method still needs a single demonstration, as Self-Replay can autonomously gather the data it needs without human supervision or interventions.
>
> > [...] some practical situations [...]
>
> During this review period, we added a new set of experiments aimed at measuring the **robustness of our method to common visual distractors** that may occur in everyday tasks, like **unseen objects** in the scene or **lighting changes**. We demonstrated with new videos on our website (https://sites.google.com/view/self-replay) and a new series of experiments in the Supplementary Material (https://bit.ly/3ksLAYo, Table 3, page 13) that our perception network, a Deep Spatial Autoencoder [3],  can extract keypoints that are robust to distactors. These keypoints autonomously learn to track the objects used in the task, and ignore novel objects added as distractors at test time. We saw how task-solving performance does not drop considerably when adding distractors.
>
> We believe these series of additional experiments and insights we provided during this review phase convincingly show the strengths of our method, its ability to solve everyday-like tasks with one demonstration on a real robot, while being robust to common sources of visual change. Despite only observing one configuration of the objects at training time, **our method can solve novel, unseen configurations, even in the presence of distractors.**
>
> [1] Lynch, Corey, et al. "Learning latent plans from play." Conference on Robot Learning. PMLR, 2020.
>
> [2] Abhishek Gupta, Vikash Kumar, Corey Lynch, Sergey Levine, Karol Hausman. Relay Policy Learning: Solving Long-Horizon Tasks via Imitation and Reinforcement Learning
>
> [3] Chelsea Finn, Xin Yu Tan, Yan Duan, Trevor Darrell, Sergey Levine, Pieter Abbeel, Deep Spatial Autoencoders for Visuomotor Learning

---

> > ### Author Response · Authors · 2021-08-30
> > **Clarification: post was written by the authors**
> >
> > We are adding this short response to clarify that the above message was written by us, the authors of the paper. Despite being logged in, the message was published as "Anonymous".

---

### Decision · Program_Chairs · 2021-09-13

**Decision:**

Accept (Poster)

**Comment:**

In this paper, the authors propose a method to solve multi-stage task from one demonstration. Multiple bottleneck states are defined and recognized. It is well recognized that the system is able to collect scalable data in a self-supervised fashion. However, there are some concerns about how to handle multiple demonstrations, some practical situations, etc. Also, more comparison experiments with stronger baselines are lacking.

In the rebuttal session, the authors have responded to the reviewers' concerns and more details and baselines are provided. And the reviewers have reached a consensus of accepting this paper.